# Conformal Anomaly Detection in Event Sequences

**Shuai Zhang** [1]   **Chuan Zhou** [1][2]   **Yang Liu** [1]   **Peng Zhang** [3]   **Xixun Lin** [4]   **Shirui Pan** [5]

## Abstract

Anomaly detection in continuous-time event sequences is a crucial task in safety-critical applications. While existing methods primarily focus on developing a superior test statistic, they fail to provide guarantees regarding the false positive rate (FPR), which undermines their reliability in practical deployments. In this paper, we propose CADES (Conformal Anomaly Detection in Event Sequences), a novel test procedure based on conformal inference for the studied task with finite-sample FPR control. Specifically, by using the time-rescaling theorem, we design two powerful non-conformity scores tailored to event sequences, which exhibit complementary sensitivities to different abnormal patterns. CADES combines these scores with Bonferroni correction to leverage their respective strengths and addresses non-identifiability issues of existing methods. Theoretically, we prove the validity of CADES and further provide strong guarantees on calibration-conditional FPR control. Experimental results on synthetic and real-world datasets, covering various types of anomalies, demonstrate that CADES outperforms state-of-the-art methods while maintaining FPR control.

## 1. Introduction

Event sequences consist of the timestamps and marks of discrete events occurring in continuous time. They are abundant and ubiquitous in our daily life. Examples include news dissemination on social networks (Trivedi et al., 2019), trading actions in stock markets (Zuo et al., 2020), and electronic health records in medical systems (Shi et al., 2023).

Detecting anomalies in event data plays a vital role in safety-critical applications, including information security, finance, and healthcare. For example, rapid information spreading may signal rumors (Naumzik & Feuerriegel, 2022), abnormal transaction activities may reveal fraud (Zhu et al., 2020), and irregular patient health records may suggest rare medical conditions (Liu & Hauskrecht, 2021).

Continuous-time event sequences are commonly modeled using temporal point processes (TPPs), a class of stochastic processes that characterize the random occurrence of events (Zhang et al., 2020). Given access to normal training data, i.e., in-distribution (ID) data, previous works (Shchur et al., 2021; Zhang et al., 2023) frame the task of detecting anomalous event sequences as an out-of-distribution (OOD) detection problem for TPPs. They perform hypothesis testing to determine whether an event sequence is drawn from the unknown data-generating TPP. While these studies primarily focus on developing a superior test statistic, they lack rigorous theoretical guarantees for controlling the false positive rate (FPR, i.e., the probability of misclassifying an ID sample as OOD). Such guarantees, however, are crucial for the deployment of OOD detection methods in safety-critical applications (Kaur et al., 2022; Magesh et al., 2023).

Conformal inference (a.k.a. conformal prediction, Vovk et al. (2005)) provides a flexible framework for establishing the desired FPR guarantees. Its application in OOD detection is often referred to as conformal OOD detection (Kaur et al., 2022) or conformal anomaly detection (Smith et al., 2015). The detection performance largely depends on the choice of the non-conformity score[1] (Kaur et al., 2022; Angelopoulos et al., 2024), which quantifies how different an input is from the ID samples. Recent advances in this field have received significant attention (Bates et al., 2023; Ma et al., 2024; Marandon et al., 2024), but these approaches have mainly focused on image data, leaving continuous-time event sequence data unexplored.

In this paper, we aim to extend conformal inference to anomaly detection in event sequences. Common test statistics for event sequences, such as the Kolmogorov-Smirnov (KS) statistic (Barnard, 1953) and the sum-of-squared-spacings statistic (Shchur et al., 2021), suffer from severe

---

[1]Academy of Mathematics and Systems Science, Chinese Academy of Sciences [2]School of Cyber Security, University of Chinese Academy of Sciences [3]Cyberspace Institute of Advanced Technology, Guangzhou University [4]Institute of Information Engineering, Chinese Academy of Sciences [5]Griffith University. Correspondence to: Chuan Zhou <zhouchuan@amss.ac.cn>.

*Proceedings of the 42$^{nd}$ International Conference on Machine Learning*, Vancouver, Canada. PMLR 267, 2025. Copyright 2025 by the author(s).

---

[1]In this paper, we use the terms *non-conformity score* and *test statistic* interchangeably.

non-identifiability issues (Zhang et al., 2024b), producing similar values for two markedly distinct sequences. This significantly reduces their detection power, motivating us to design a more expressive non-conformity score. Due to the variable length of event sequences and the intricate dependencies between events, the design of this score is both essential and non-trivial.

To this end, we introduce CADES (Conformal Anomaly Detection in Event Sequences), a novel method based on conformal inference for detecting anomalous event sequences. Specifically, we design two powerful non-conformity scores tailored to event sequences using the time-rescaling theorem (Brown et al., 2002). These scores are complementary in their sensitivity to different abnormal patterns and address the limitations mentioned above of existing test statistics. Unlike standard conformal inference methods, which rely on one-sided p-values and a single non-conformity score, CADES utilizes two-sided p-values and applies the Bonferroni correction (Shaffer, 1995) to combine both scores. This is because two-sided p-values are crucial, as we show that both small and large values of the proposed scores can indicate OOD sequences. Moreover, by combining both scores, CADES fully exploits the strengths of each, enabling more accurate anomaly detection. Theoretically, we prove the validity of CADES and provide guarantees on calibration-conditional FPR, which are stronger than marginal FPR guarantees averaged over all possible calibration sets.

We summarize our contributions as follows:

- We establish the connection between conformal inference and anomaly detection in event sequences. Building upon this, we propose CADES, a novel test procedure that combines two proposed scores with Bonferroni correction to perform hypothesis testing, offering a statistically sound anomaly detection method.

- We design two new powerful non-conformity scores tailored to continuous-time event sequences. We demonstrate that these scores complement each other in terms of sensitivity to different anomalies and address non-identifiability issues of existing test statistics.

- We prove the validity of the p-values used in CADES. This guarantees controlling the marginal FPR at a pre-specified level. We further provide theoretical guarantees on calibration-conditional FPR, which are stronger than marginal FPR guarantees.

- We conduct extensive experiments on synthetic and real-world datasets, covering various types of anomalies, to validate the effectiveness of CADES[2]. The results show that CADES outperforms state-of-the-art methods with respect to both detection performance and FPR control.

[2]The code is publicly available at https://github.com/Zh-Shuai/CADES.

## 2. Preliminaries

**Temporal Point Processes.** A temporal point process (TPP, Daley et al. (2003)) is a stochastic process whose realization is an event sequence $X = \{(t_i, m_i)\}_{i=1}^{N}$, where $t_i \in [0, T]$ are the event timestamps with $t_i < t_{i+1}$, and $m_i \in \mathcal{M} = \{1, \ldots, M\}$ are the event types (marks). It can be equivalently represented as $M$ counting processes $\{N_m(t)\}_{m=1}^{M}$, where $N_m(t)$ denotes the number of type-$m$ events occurring up to time $t$. The most common way to characterize a TPP is through conditional intensity functions (CIFs), defined for each event type $m \in \mathcal{M}$ as follows:

$$\lambda_m^*(t)\, \mathrm{d}t = \mathbb{E}\left[\mathrm{d}N_m(t) \mid \mathcal{H}_t^{\mathcal{M}}\right], \quad (1)$$

where $\mathcal{H}_t^{\mathcal{M}} = \{(t_i, m_i) \mid t_i < t, m_i \in \mathcal{M}\}$ represents the historical events of all types before time $t$, and the symbol $*$ indicates that the intensity is conditioned on $\mathcal{H}_t^{\mathcal{M}}$. This CIF describes the expected instantaneous rate of happening next type-$m$ event given historical events.

Traditional TPP models include Poisson processes (Kingman, 1992), Hawkes processes (Hawkes, 1971), and self-correcting processes (Isham & Westcott, 1979). Recently, neural TPPs, which parameterize the CIFs using neural networks such as RNNs or Transformers, have gained significant attention (Mei & Eisner, 2017; Zuo et al., 2020).

**Conformal Anomaly Detection.** We focus on the computationally efficient inductive (split) conformal anomaly detection method (Laxhammar & Falkman, 2015), which divides the clean training dataset $\mathcal{D}$ into two disjoint subsets: a proper training set $\mathcal{D}_{\mathrm{train}}$ and a calibration set $\mathcal{D}_{\mathrm{cal}}$. First, a model is trained on $\mathcal{D}_{\mathrm{train}}$ to define a non-conformity score $s : \mathcal{X} \to \mathbb{R}$. The calibration scores $\{s(X_{\mathrm{cal}}) : X_{\mathrm{cal}} \in \mathcal{D}_{\mathrm{cal}}\}$ and the test score $s(X_{\mathrm{test}})$ are then calculated. Next, the classical conformal p-value (Vovk et al., 2005) for the test sample $X_{\mathrm{test}}$ is given by:

$$p(X_{\mathrm{test}}) = \frac{|\{X_{\mathrm{cal}} \in \mathcal{D}_{\mathrm{cal}} : s(X_{\mathrm{test}}) \leq s(X_{\mathrm{cal}})\}| + 1}{|\mathcal{D}_{\mathrm{cal}}| + 1}. \quad (2)$$

If the elements of $\mathcal{D}_{\mathrm{cal}} \cup \{X_{\mathrm{test}}\}$ are exchangeable, i.e., their joint distribution is invariant under any permutation, then the conformal p-value is valid, ensuring that the FPR is controlled (Papadopoulos et al., 2002). In this case, if $p(X_{\mathrm{test}})$ is smaller than a pre-specified significance level $\alpha \in (0, 1)$, $X_{\mathrm{test}}$ is classified as an anomaly.

**Problem Statement.** Assuming that normal training data is available, detecting anomalous event sequences can be viewed as an OOD detection problem. Concretely, given a set of event sequences $\mathcal{D} = \{X_i\}_{i=1}^{n}$ that are i.i.d. samples from an unknown distribution $P_X$, the objective is to identify whether a new independent observation $X_{\mathrm{test}}$ is OOD, in the sense that it was not drawn from $P_X$. This problem can be framed as a hypothesis testing for TPPs:

$$H_0 : X_{\mathrm{test}} \sim P_X \quad \text{vs.} \quad H_1 : X_{\mathrm{test}} \nsim P_X. \quad (3)$$

Here we consider $P_X$ to be some unknown data-generating TPP. If the null hypothesis $H_0$ is rejected, we declare that $X_{\text{test}}$ is OOD or abnormal; otherwise, it is considered ID or normal. In this paper, our goal is to develop a test procedure that can identify true anomalies as accurately as possible, while controlling the FPR at a pre-specified level.

## 3. Proposed Method: CADES

In this section, we present CADES, a novel test procedure based on conformal inference for detecting anomalous event sequences. We begin by introducing the concept of the valid p-value (Casella & Berger, 2024), which ensures control over the FPR as specified by Eq.(4).

**Definition 3.1.** A p-value $p(X)$ is a statistic satisfying $0 \leq p(X) \leq 1$ for every sample $X$. Small values of $p(X)$ give evidence against the null hypothesis $H_0$. A p-value is valid if, for any $\alpha \in (0, 1)$,

$$\mathbb{P}_{H_0}(p(X) \leq \alpha) \leq \alpha. \tag{4}$$

Typically, the p-value is the probability under the null hypothesis of obtaining a real-valued test statistic at least as extreme as the one observed (Haroush et al., 2022). However, in the OOD detection problem (3), the null distribution $P_X$ is unknown, making it infeasible to compute the exact p-values. Inspired by conformal inference (Vovk et al., 2005), we estimate p-values using conformal p-values. Before this, it is necessary to design a non-conformity score (i.e. test statistic) to quantify how different an observed sequence is from the training (ID) sequences. The design of this score is a key factor for detection performance (Angelopoulos et al., 2024): different designs can lead to very different results, and a poorly designed score can lead to completely ineffective detection, e.g., with extremely low detection power. Although many expressive non-conformity scores have been developed for image data (Bates et al., 2023; Ma et al., 2024; Marandon et al., 2024), their exploration for event sequences remains limited.

In what follows, we discuss the limitations of existing test statistics for event sequences and elaborate on the definition of our two non-conformity scores in Section 3.1. Section 3.2 details our test procedure, CADES, which combines the two proposed scores for OOD detection. We prove the validity of the p-value used in CADES and provide theoretical guarantees on calibration-conditional FPR in Section 3.3.

### 3.1. Non-Conformity Scores for Event Sequences

A natural way to score an event sequence is to compute its negative log-likelihood (NLL) under a trained neural TPP model. However, Shchur et al. (2021) demonstrated that the NLL statistic is less effective than their proposed sum-of-squared-spacings (3S) statistic for detecting anomalous event sequences. Other widely used statistics include Kolmogorov-Smirnov (KS) statistics, which are popular in the goodness-of-fit (GOF) test for TPPs (Barnard, 1953; Gerhard & Gerstner, 2010; Li et al., 2018). Nevertheless, KS statistics lack sensitivity to the event count and suffer from severe non-identifiability issues, producing similar values for two markedly distinct event sequences (Shchur et al., 2021; Zhang et al., 2024b). Similarly, the 3S statistic, along with the $Q_+$ and $Q_-$ statistics (Zhang et al., 2023), are insensitive to relatively uniform spacings (i.e. inter-event times), which significantly reduces their detection power.

The above limitations highlight the urgent need for a more expressive non-conformity score that can accurately quantify the discrepancy between an input sequence and ID sequences. Different from traditional settings, where each subject is characterized by a fixed number of features, an event sequence $X = \{(t_i, m_i)\}_{i=1}^N$ on $[0, T]$ varies in length. Moreover, events are associated with diverse marks and intricate dependencies exist among them. Therefore, designing a reasonable and expressive score $s(X)$ that accounts for these factors is a crucial yet non-trivial task. We draw inspiration from the following time-rescaling theorem (adapted from Daley et al. (2003)) to transform the event sequence and lay the foundation for a novel scoring mechanism.

**Theorem 3.2.** *Let $X = \{(t_i, m_i)\}_{i=1}^N$ be a sequence of random event points on the interval $[0, T]$ corresponding to a TPP $\{N_m(t)\}_{m=1}^M$ with CIFs $\{\lambda_m^*(t)\}_{m=1}^M$. For each $m \in \mathcal{M}$, denote the events of type-$m$ from $N_m(t)$ as $X^{(m)} = \left(t_1^{(m)}, \ldots, t_{N_m(T)}^{(m)}\right)$, where the number of events satisfies $\sum_{m=1}^M N_m(T) = N$. If each $\lambda_m^*(t)$ is positive on $[0, T]$ and $\Lambda_m^*(T) = \int_0^T \lambda_m^*(s)\,\mathrm{d}s < \infty$ almost surely, then for each $m \in \mathcal{M}$, the transformed sequence*

$$Z^{(m)} = \left(\Lambda_m^*\left(t_1^{(m)}\right), \ldots, \Lambda_m^*\left(t_{N_m(T)}^{(m)}\right)\right) \tag{5}$$

*forms a Poisson process with unit rate on $[0, \Lambda_m^*(T)]$. Moreover, the sequences $\left\{Z^{(m)}\right\}_{m=1}^M$ are independent.*

This theorem states that by using the integrated CIFs, an event sequence $X$ can be transformed into $M$ independent realizations $\{Z^{(m)}\}_{m=1}^M$ of the unit-rate Poisson process (a.k.a. the standard Poisson process, SPP). Following prior work (Shchur et al., 2021), we concatenate the rescaled sequences $\{Z^{(m)}\}_{m=1}^M$ for each mark into a single SPP realization $Z$ on the interval $\left[0, \sum_{m=1}^M \Lambda_m^*(T)\right]$. Specifically, $Z$ is expressed as:

$$Z = \left(\widetilde{Z^{(1)}} \,\|\, \cdots \,\|\, \widetilde{Z^{(m)}} \,\|\, \cdots \,\|\, \widetilde{Z^{(M)}}\right), \tag{6}$$

where the symbol $\|$ denotes vector concatenation, $\widetilde{Z^{(m)}} := V_m + Z^{(m)} = \left(V_m + \Lambda_m^*\left(t_1^{(m)}\right), \ldots, V_m + \Lambda_m^*\left(t_{N_m(T)}^{(m)}\right)\right)$, and $V_m = \sum_{i=1}^{m-1} \Lambda_i^*(T)$, for each $m \in \mathcal{M}$.

For simplicity, we rewrite $Z = (\tau_1, \ldots, \tau_N)$ and denote its observation length as $V = \sum_{m=1}^{M} \Lambda_m^*(T)$. We adopt the inductive conformal inference method (Vovk et al., 2005), where the dataset $\mathcal{D}$ of size $n$ is randomly divided into two disjoint subsets: a proper training set $\mathcal{D}_{\text{train}}$ of size $n_{\text{train}} < n$ and a calibration set $\mathcal{D}_{\text{cal}}$ of size $n_{\text{cal}} = n - n_{\text{train}}$. To design a non-conformity score for an event sequence $X$, we first rescale $X$ to obtain $Z$ using a neural TPP model trained on $\mathcal{D}_{\text{train}}$, and then quantify how different the transformed sequence $Z$ is from the SPP on the interval $[0, V]$. Due to the powerful capability of neural networks, the neural TPP model provides a more accurate approximation of the true CIFs of training sequences.

One property of the SPP is that, conditionally on the event count in $[0, V]$ being equal to $N$, the normalized arrival times $\tau_1/V, \ldots, \tau_N/V$ are independently and uniformly distributed on $[0, 1]$ (Lewis, 1965; Cox & Lewis, 1966). Building on this property, we propose a non-conformity score using the Kullback-Leibler (KL) divergence to quantify the discrepancy between the underlying distribution of the normalized arrival times and the $\text{Uniform}([0, 1])$ distribution. In the literature, KL divergence is a natural measure of the difference between two probability distributions, making it an appropriate choice for our non-conformity score. Since the continuous version of KL divergence is defined on probability density functions (PDFs), we apply kernel density estimation (KDE, Scott (2015)) with the Gaussian kernel to estimate the underlying PDF of the normalized arrival times. Consequently, the proposed score for an event sequence $X$ is formally defined as follows:

$$s_{\text{arr}}(X) := D_{\text{KL}}(f_{\text{arr}} \| \hat{f}_{\text{arr}}) = \int_{-\infty}^{\infty} f_{\text{arr}}(x) \log \left( \frac{f_{\text{arr}}(x)}{\hat{f}_{\text{arr}}(x)} \right) \mathrm{d}x,$$
(7)

where $f_{\text{arr}}(x) = \mathbb{1}_{[0,1]}(x)$ is the uniform PDF, and $\hat{f}_{\text{arr}}(x) = \frac{1}{h_1 N} \sum_{i=1}^{N} \phi(\frac{x - \tau_i/V}{h_1})$ is the KDE of the normalized arrival times $\tau_i/V$ of $Z$, obtained by applying time-rescaling and concatenation to $X$. $h_1 > 0$ is a parameter called the bandwidth, and $\phi(\cdot)$ denotes the standard normal PDF. We compute the integral using numerical techniques, such as the trapezoidal rule (Stoer et al., 1980).

It can be observed that $s_{\text{arr}}$ exhibits sensitivity to reductions in event count, effectively overcoming a major limitation of KS statistics. For example, if all events of $Z$ in the subinterval $[\frac{V}{2}, V]$ are removed, then the remaining normalized arrival times $\tau_i/V$ will be confined to $[0, \frac{1}{2}]$, resulting in a pronounced deviation between the KDE $\hat{f}_{\text{arr}}$ and the uniform PDF. Nevertheless, by definition, $s_{\text{arr}}$ is less sensitive to relatively uniform arrival times, a limitation also shared by the 3S statistic as well as the $Q_+$ and $Q_-$ statistics.

To address this issue and further utilize the information of the SPP, we introduce another non-conformity score, $s_{\text{int}}$,

based on the fact that the inter-event times $w_i = \tau_i - \tau_{i-1}$ in the SPP follow an $\text{Exponential}(1)$ distribution (Cox & Lewis, 1966). Specifically, $s_{\text{int}}$ is defined as:

$$s_{\text{int}}(X) := D_{\text{KL}}(f_{\text{int}} \| \hat{f}_{\text{int}}) = \int_{-\infty}^{\infty} f_{\text{int}}(x) \log \left( \frac{f_{\text{int}}(x)}{\hat{f}_{\text{int}}(x)} \right) \mathrm{d}x,$$
(8)

where $f_{\text{int}}(x) = \mathrm{e}^{-x} \mathbb{1}_{[0,\infty)}(x)$ is the exponential PDF, and $\hat{f}_{\text{int}}(x) = \frac{1}{h_2(N+1)} \sum_{i=1}^{N+1} \phi(\frac{x - w_i}{h_2})$ is the KDE of the inter-event times $w_i = \tau_i - \tau_{i-1}$, with $\tau_0 = 0$ and $\tau_{N+1} = V$. $h_2 > 0$ is another bandwidth parameter.

We find that $s_{\text{int}}$ is more sensitive to relatively uniform arrival times compared to $s_{\text{arr}}$. For example, when all arrival times $\tau_i$ and the observed length $V$ lie near integer grid points, the inter-event times $w_i$ are close to 1. This causes the KDE $\hat{f}_{\text{int}}$ to concentrate around 1, making it clearly distinguishable from the exponential PDF. In contrast, $s_{\text{int}}$ is less sensitive than $s_{\text{arr}}$ to reductions in event count. As a result, $s_{\text{arr}}$ and $s_{\text{int}}$ offer complementary sensitivities to different types of anomalies. To fully exploit their respective strengths, we propose combining these two scores for OOD detection, as explained in the next subsection.

### 3.2. Test Procedure with Bonferroni Correction

With the proposed score $s_{\text{arr}}$, the classical conformal p-value for the test sequence $X_{\text{test}}$ is calculated as follows:

$$p_{\text{arr}}^r(X_{\text{test}}) = \frac{|\{X_{\text{cal}} \in \mathcal{D}_{\text{cal}} : s_{\text{arr}}(X_{\text{test}}) \leq s_{\text{arr}}(X_{\text{cal}})\}| + 1}{n_{\text{cal}} + 1}.$$
(9)

This p-value is right-sided because a larger score $s_{\text{arr}}$ typically suggests that a sequence is OOD, as a large KL divergence provides evidence of the difference between two distributions. However, in certain alternative scenarios, such as arrival times being near integer grid points or the self-correcting process generating more evenly-spaced events than the SPP, the KDE $\hat{f}_{\text{arr}}$ becomes closer to the uniform PDF. In these cases, small values of $s_{\text{arr}}$ may also indicate OOD sequences. Thus, we utilize the two-sided p-value:

$$p_{\text{arr}}(X_{\text{test}}) = 2 \min\{p_{\text{arr}}^l(X_{\text{test}}), \, p_{\text{arr}}^r(X_{\text{test}})\},$$
(10)

where the left-sided p-value $p_{\text{arr}}^l(X_{\text{test}})$ is defined similarly to the above $p_{\text{arr}}^r(X_{\text{test}})$, except that the inequality in Eq.(9) is reversed. Throughout this paper, p-values such as $p_{\text{arr}}(X_{\text{test}})$ are truncated at 1 whenever they exceed it.

For another score $s_{\text{int}}$, the two-sided p-value, denoted as $p_{\text{int}}(X_{\text{test}})$, can be calculated in the same way. To leverage the complementary sensitivities of $s_{\text{arr}}$ and $s_{\text{int}}$ to different abnormal patterns, we combine them for OOD detection using the Bonferroni corrected p-value (Shaffer, 1995):

$$p_{\text{cor}}(X_{\text{test}}) = \min\{2(1+\varepsilon)p_{\text{arr}}(X_{\text{test}}), 2(1+\varepsilon)p_{\text{int}}(X_{\text{test}})\},$$
(11)

where $\varepsilon \geq 0$ is a parameter. Following Magesh et al. (2023), we introduce the factor $(1 + \varepsilon)$ in Eq.(11) to provide strong false positive guarantees conditioned on the calibration set, as detailed in Theorem 3.4. The value of $\varepsilon$ depends on the calibration set size $n_{\text{cal}}$: a smaller $\varepsilon$ improves detection power but requires a larger $n_{\text{cal}}$ to maintain these guarantees. Conversely, when $n_{\text{cal}}$ is small, a larger $\varepsilon$ is needed to ensure the guarantees, making the detection more conservative.

For a given significance level $\alpha \in (0, 1)$, we declare the test sequence $X_{\text{test}}$ as OOD if $p_{\text{cor}}(X_{\text{test}}) \leq \alpha$. This ensures a more accurate detection of anomalies by identifying discrepancies in the normalized arrival times $\tau_i/V$ relative to the uniform distribution and in the inter-event times $w_i$ relative to the exponential distribution. We summarize our test procedure in Algorithm 1 and provide rigorous theoretical guarantees subsequently.

### 3.3. Theoretical Guarantees

We first prove the validity of the p-value used in our CADES method, which ensures control of the marginal FPR. Building upon this, we then demonstrate that, under certain conditions, our method also controls the calibration-conditional FPR with high probability.

It can be verified that the p-value $p_{\text{cor}}(X_{\text{test}})$ in Eq.(11) remains super-uniform and, therefore, a valid p-value. We state this result formally in the following proposition:

**Proposition 3.3.** *Suppose the test sequence $X_{\text{test}}$ and the dataset $\mathcal{D}$ are i.i.d., then the p-value $p_{\text{cor}}(X_{\text{test}})$ is valid, i.e., for every $\alpha \in (0, 1)$, $\mathbb{P}_{H_0}(p_{\text{cor}}(X_{\text{test}}) \leq \alpha) \leq \alpha$.*

The proof can be found in Appendix B.1. In deed, $p_{\text{cor}}(X_{\text{test}})$ is marginally valid because it depends on the calibration set $\mathcal{D}_{\text{cal}}$. Since $\mathbb{P}_{H_0}(\text{declare OOD}) = \mathbb{P}_{H_0}(p_{\text{cor}}(X_{\text{test}}) \leq \alpha) \leq \alpha$, we obtain:

$$\mathbb{P}_{H_0}(\text{declare OOD}) = \mathbb{E}\left[\mathbb{P}_{H_0}(\text{declare OOD} \mid \mathcal{D}_{\text{cal}})\right] \leq \alpha, \tag{12}$$

where the expectation is over $\mathcal{D}_{\text{cal}}$. This indicates that the marginal FPR $\big(\text{i.e. the probability } \mathbb{P}_{H_0}(\text{declare OOD})\big)$ control is averaged over all possible calibration sets. However, it does not guarantee that the target level $\alpha$ is maintained for the particular calibration set used.

Inspired by (Magesh et al., 2023; Bates et al., 2023), we further provide stronger guarantees that Algorithm 1 can control the calibration-conditional FPR $\big(\text{i.e. the conditional probability } \mathbb{P}_{H_0}(\text{declare OOD} \mid \mathcal{D}_{\text{cal}})\big)$ with high probability, under certain conditions on the size of the calibration set.

**Theorem 3.4.** *Let $\alpha, \delta \in (0, 1)$ and $\varepsilon \geq 0$. Let $\mathcal{D}_{\text{cal}}$ be a calibration set of size $n_{\text{cal}}$, $a = \lfloor (n_{\text{cal}} + 1) \frac{\alpha}{4(1+\varepsilon)} \rfloor$, $b = n_{\text{cal}} + 1 - a$, and $\mu = \frac{a}{a+b}$. For a given $\delta > 0$, let $n_{\text{cal}}$ be such that*

$$I_{(1+\varepsilon)\mu}(a, b) \geq 1 - \frac{\delta}{4}, \tag{13}$$

---

**Algorithm 1** CADES: Conformal Anomaly Detection in Event Sequences

**Input:** Clean dataset $\mathcal{D} = \mathcal{D}_{\text{train}} \cup \mathcal{D}_{\text{cal}}$, test sequence $X_{\text{test}}$, target significance level $\alpha \in (0, 1)$.

1: Train a neural TPP with CIFs $\{\lambda_m^*(t)\}_{m=1}^M$ on $\mathcal{D}_{\text{train}}$;
2: Apply time-rescaling and concatenation for $\mathcal{D}_{\text{cal}}$ and $X_{\text{test}}$;
3: Calculate the scores $s_{\text{arr}}$ (7) and $s_{\text{int}}$ (8) for $\mathcal{D}_{\text{cal}}$ and $X_{\text{test}}$;
4: Compute the Bonferroni corrected p-value $p_{\text{cor}}(X_{\text{test}})$(11).

**Output:** Declare $X_{\text{test}}$ as OOD or abnormal if $p_{\text{cor}}(X_{\text{test}}) \leq \alpha$; otherwise, conclude it is ID or normal.

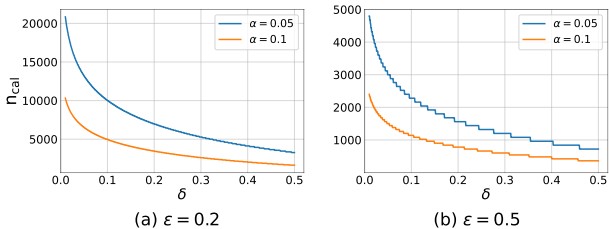

*Figure 1.* Calibration set size $n_{\text{cal}}$ that guarantees the calibration-conditional FPR is bounded by $\alpha$ with probability $1 - \delta$.

where $I_x(a, b)$ *denotes the CDF of the* $\text{Beta}(a, b)$ *distribution. If* $s_{\text{arr}}(X)$ *and* $s_{\text{int}}(X)$ *are continuously distributed, then for a new sequence* $X_{\text{test}}$*, the probability of incorrectly identifying* $X_{\text{test}}$ *as OOD conditioned on* $\mathcal{D}_{\text{cal}}$ *while using Algorithm 1 is bounded by* $\alpha$ *with probability* $1 - \delta$*, i.e.,*

$$\mathbb{P}\big[\mathbb{P}_{H_0}(\text{declare OOD} \mid \mathcal{D}_{\text{cal}}) \leq \alpha\big] \geq 1 - \delta. \tag{14}$$

We present a detailed proof in Appendix B.2. This theorem builds upon a result from (Vovk, 2012; Bates et al., 2023), which states that the CDF of the conformal p-value conditioned on the calibration set follows a Beta distribution. Additionally, due to symmetry, this property holds for both the left-sided and right-sided conformal p-values. Considering the form of the Beta distribution's CDF, it is difficult to express the dependence of $n_{\text{cal}}$ on $\alpha, \delta$ and $\varepsilon$ in closed form. We show the calibration set size $n_{\text{cal}}$ given by (13) when $\varepsilon = 0.2$ and $\varepsilon = 0.5$ for different values of $\delta$ in Figure 1.

**Remark.** The p-values in this paper, such as $p_{\text{arr}}^r(X_{\text{test}})$ in Eq.(9), are evaluated on the sequences from the calibration set $\mathcal{D}_{\text{cal}}$, rather than the entire dataset $\mathcal{D}$, as used in previous works (Shchur et al., 2021; Zhang et al., 2023). Using the entire dataset $\mathcal{D}$ may lead to invalid p-values because $s_{\text{arr}}(X_{\text{test}})$ is not exchangeable with $\{s_{\text{arr}}(X) : X \in \mathcal{D}\}$. The problem is that $s_{\text{arr}}$ has already depended on $\mathcal{D}_{\text{train}}$ (since it was defined based on a neural TPP model trained on $\mathcal{D}_{\text{train}}$), but it has not yet seen the test sample $X_{\text{test}}$ (Tibshirani, 2023). In contrast, using the hold-out calibration set $\mathcal{D}_{\text{cal}}$ preserves the exchangeability of $s_{\text{arr}}(X_{\text{test}})$ and $\{s_{\text{arr}}(X_{\text{cal}}) : X_{\text{cal}} \in \mathcal{D}_{\text{cal}}\}$, ensuring the validity of p-values (Papadopoulos et al., 2002).

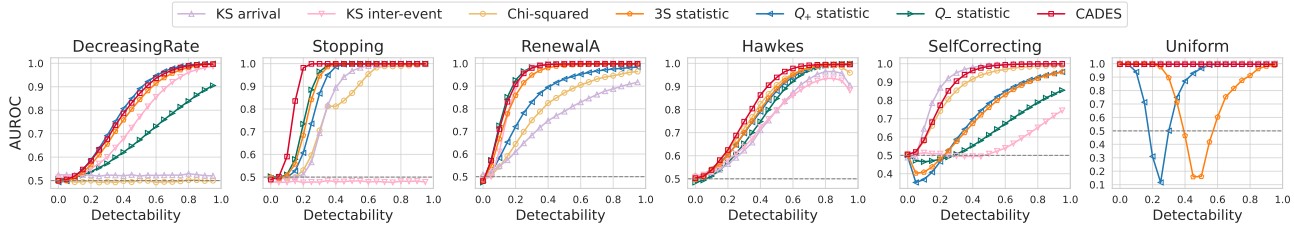

*Figure 2.* Performance of GOF test for the standard Poisson process measured by AUROC (higher is better).

# 4. Experiments

In this section, we conduct extensive experiments to evaluate our CADES method, including GOF test for SPP (Section 4.1), anomaly detection in synthetic data (Section 4.2) and real-world data (Section 4.3), and FPR control (Section 4.4). In addition, we perform ablation studies to verify the effectiveness of combining two proposed scores and using two-sided p-values in Section 4.5. Appendix D contains runtime comparisons and additional experimental results.

## 4.1. GOF Tests for SPP under Nine Alternatives

Existing goodness-of-fit (GOF) statistics for the standard Poisson process (SPP) are either insensitive to the event count or to relatively uniform spacings. As described in Section 3.1, the two proposed non-conformity scores can overcome these limitations. Hence, in this subsection, we empirically evaluate the performance of the GOF test for the SPP using the combination of these two scores.

**Datasets.** ID sequences and OOD sequences are generated from the SPP and the alternative distribution on the interval $[0, 100]$, respectively. We consider nine choices for the alternative distribution: (1) DecreasingRate, (2) IncreasingRate, (3) InhomogeneousPoisson, (4) Stopping, (5) RenewalA, (6) RenewalB, (7) Hawkes, (8) SelfCorrecting, and (9) Uniform. For each alternative distribution, a detectability parameter $\eta \in [0, 1]$ is defined. The first eight scenarios follow previous work (Shchur et al., 2021), where a higher value of $\eta$ indicates a greater dissimilarity of the alternative distribution from the SPP. In the Uniform scenario, we generate the OOD sequence with equal inter-event times. For all scenarios, $\mathcal{D}$ consists of 1000 ID sequences, while $\mathcal{D}_{\text{test}}^{\text{ID}}$ and $\mathcal{D}_{\text{test}}^{\text{OOD}}$ consist of 1000 ID sequences and 1000 OOD sequences, respectively. Detailed descriptions of these alternative distributions are provided in Appendix C.1.1.

**Baselines.** We consider six test statistics: (1) KS statistic on arrival times, (2) KS statistic on inter-event times, (3) Chi-squared statistic on arrival times (Cox, 1955), (4) 3S statistic (Shchur et al., 2021), (5) $Q_+$ statistic (Zhang et al., 2023), and (6) $Q_-$ statistic (Zhang et al., 2023). See Appendix C.2 for the definitions of these statistics. For each statistic, the two-sided p-value for the test sequence is calculated with its empirical distribution function (EDF) on the ID dataset $\mathcal{D}$.

**Experimental Setup.** For this GOF test, training a neural TPP model is unnecessary, since the distribution of the null hypothesis (i.e. the SPP) is known. Hence, we treat the entire dataset $\mathcal{D}$ as the calibration set. Moreover, time-rescaling and concatenation operations are also not required. We perform the GOF test following steps 3 and 4 in Algorithm 1. Due to the limited sample size in the benchmark dataset, ensuring calibration-conditional FPR control would require a large $\varepsilon$ in Eq.(11), which could make the detection overly conservative. Therefore, in the experiments, we use the marginally valid p-value $p_{\text{cor}}(X_{\text{test}})$ and set $\varepsilon = 0$ to improve detection power. The bandwidth values for $s_{\text{arr}}(X)$ and $s_{\text{int}}(X)$ are determined by a grid search. More details of the implementation can be found in Appendix C.4.

**Evaluation Metric.** In line with previous works (Shchur et al., 2021; Zhang et al., 2023), we evaluate the performance of distinguishing ID and OOD sequences by the area under the receiver operating characteristic curve (AUROC).

**Results.** The AUROC scores across six scenarios with varying detectability parameters are shown in Figure 2, while the results for the remaining three scenarios are detailed in Appendix D.2 due to space constraints. The results indicate that KS arrival, KS inter-event, and Chi-squared statistics lose their detection capability when the number of events changes substantially, as observed in the DecreasingRate and Stopping scenarios. The 3S statistic and the $Q_+$ and $Q_-$ statistics perform poorly in scenarios with relatively uniform spacings, such as the SelfCorrecting and Uniform cases. Specifically, we explain below why the 3S statistic does not perform well under the Uniform scenario. According to Proposition 1 in (Shchur et al., 2021), the value of the 3S statistic for an SPP realization (i.e. an ID sequence) is around 2. In the Uniform scenario, the spacings of the OOD sequences are identical. For example, when the detectability parameter $\eta = 0.5$, the spacings are all 2. In this case, the 3S statistic for an OOD sequence is equal to 2, making it unable to distinguish between ID and OOD sequences.

In contrast, our CADES method consistently achieves the best or nearly the best performance across all 9 scenarios. Notably, in the Stopping case, CADES significantly outperforms baselines due to the sensitivity of the proposed score $s_{\text{arr}}$ to reductions in event count. Additional experimental analyses are provided in the ablation study in Section 4.5.

*Figure 3.* Performance of OOD detection on synthetic datasets measured by AUROC (higher is better).

*Table 1.* AUROC (%) for OOD detection on real-world datasets. Best results are in **bold** and second best are underlined.

| Dataset | KS arrival | KS inter-event | Chi-squared | Log-likelihood | 3S statistic | MultiAD-$Q_+$ | MultiAD-$Q_-$ | CADES (ours) |
|---|---|---|---|---|---|---|---|---|
| LOGS - Packet corruption (1%) | 47.24 | 71.80 | 67.27 | 90.92 | 95.03 | 92.44 | **96.61** | 96.48 |
| LOGS - Packet corruption (10%) | 64.96 | 98.72 | 49.35 | 98.98 | 99.30 | 99.31 | **99.53** | 99.48 |
| LOGS - Packet duplication (1%) | 61.88 | 79.59 | 21.26 | 81.97 | 91.46 | 91.24 | 78.15 | **92.88** |
| LOGS - Packet delay (frontend) | 90.31 | 47.46 | 95.70 | **99.55** | 96.10 | 97.97 | 95.27 | 98.15 |
| LOGS - Packet delay (all services) | 95.13 | 96.60 | 94.35 | 96.30 | 99.16 | **99.59** | 99.31 | 99.33 |
| STEAD - Anchorage, AK | 62.31 | 78.44 | 70.75 | 88.16 | 91.73 | 84.00 | 99.16 | **99.31** |
| STEAD - Aleutian Islands, AK | 53.37 | 86.48 | 64.17 | 97.08 | 99.80 | 99.86 | 99.84 | **99.95** |
| STEAD - Helmet, CA | 61.94 | 98.83 | 73.62 | 96.96 | 93.82 | 70.71 | 99.13 | **99.30** |
| Average Rank | 7.50 | 5.63 | 7.00 | 4.25 | 3.63 | 3.50 | 3.00 | **1.50** |

## 4.2. Detecting Anomalies in Synthetic Data

In this subsection, we evaluate the performance of CADES in detecting anomalous event sequences on synthetic data, which corresponds to performing OOD detection. Different from the GOF test for SPP in the previous subsection, OOD detection requires training a neural TPP model because the distribution under the null hypothesis is unknown.

**Datasets.** We consider four synthetic datasets introduced in (Shchur et al., 2021). The ServerStop, ServerOverload, and Latency datasets simulate anomaly detection in server logs, while the SpikeTrains dataset evaluates the detection of anomalies caused by event mark swaps. For each scenario, a detectability parameter $\eta$ is defined as before. See Appendix C.1.2 for dataset details.

**Baselines.** We compare our CADES method with the following baselines:

- KS arrival, KS inter-event, Chi-squared, Log-likelihood, and 3S statistic: Shchur et al. (2021) train a neural TPP on $\mathcal{D}$ and compute test statistics on the transformed sequence $Z$, including KS statistics, the Chi-squared statistic, the log-likelihood of the learned model, and their proposed 3S statistic. The two-sided p-values are calculated using the EDF of each statistic on $\mathcal{D}$. We refer to each detection method by the corresponding test statistic.

- MultiAD-$Q_+$ and MultiAD-$Q_-$: Zhang et al. (2023) train a neural TPP on $\mathcal{D}$ and conduct multiple testing on the rescaled sequence $Z^{(m)}$ for $m \in \mathcal{M}$. Two-sided p-values are calculated using the kernel cumulative distribution function of their proposed $Q_+$ or $Q_-$ statistic on $\mathcal{D}$.

**Experimental Setup.** We randomly split $\mathcal{D}$ into two disjoint subsets of equal size: a training set $\mathcal{D}_{\text{train}}$ and a calibration set $\mathcal{D}_{\text{cal}}$. Unlike the baselines that train a neural TPP on the entire dataset $\mathcal{D}$, we train it on the subset $\mathcal{D}_{\text{train}}$. We conduct OOD detection as described in Algorithm 1. To ensure a fair comparison, we employ the same neural TPP (LogNormMix, Shchur et al. (2020)) as the baselines, utilizing the identical model architecture and training procedure. Model and training details can be found in Appendix C.3.

**Results.** As shown in Figure 3, our method achieves the best or comparable performance across four synthetic datasets. In particular, CADES significantly outperforms baselines on the ServerStop and ServerOverload datasets, even when the detectability $\eta = 0.05$, where OOD sequences are obtained by changing only 5% of the time interval in ID sequences.

## 4.3. Detecting Anomalies in Real-World Data

We further evaluate CADES on two benchmark real-world datasets, LOGS and STEAD, both introduced in (Shchur et al., 2021). See Appendix C.1.3 for detailed descriptions of these datasets. The experimental setup remains identical to that in Section 4.2.

**Results.** Table 1 summarizes the experimental results. It can be observed that CADES consistently achieves the best or near-best AUROC across all scenarios, with an average rank significantly outperforming baselines. Notably, CADES demonstrates strong performance even in challenging scenarios such as "STEAD - Helmet, CA" and "LOGS - Packet duplication (1%)", where competing methods like MultiAD-$Q_+$ and MultiAD-$Q_-$ exhibit limitations, respectively.

## 4.4. FPR Control

We treat OOD sequences as *positives* and ID sequences as *negatives*. The false positive rate (FPR) control guarantees are crucial for safety-critical applications, such as fraud detection and medical diagnosis. For example, in electronic health records (EHRs), maintaining low FPR is vital. High false positives could lead to incorrect diagnoses or unnecessary treatments, potentially compromising patient safety. Our method ensures that detected anomalies are more likely to be genuine, helping healthcare professionals make more reliable decisions.

We empirically investigate whether CADES and the baselines can control the FPR. The target significance level $\alpha$ is taken from 0.05 to 0.5, with a step size of 0.05. For each $\alpha$, we conduct five repeated experiments on the real-world datasets LOGS and STEAD and present the FPR boxplots in Figure 4. As shown, the FPR of CADES is upper bounded by $\alpha$ on average. However, the baseline methods, the 3S statistic and MultiAD-$Q_+$, exhibit a much higher FPR than the pre-specified level $\alpha$ on the LOGS dataset. Since the baselines also control FPR on the STEAD dataset, we further provide their true positive rate (TPR), i.e., detection power, at the commonly used level $\alpha = 0.05$ in Table 2. We can observe that CADES significantly outperforms the baselines in identifying true anomalous sequences. These results highlight the effectiveness of CADES in controlling FPR while achieving superior detection power.

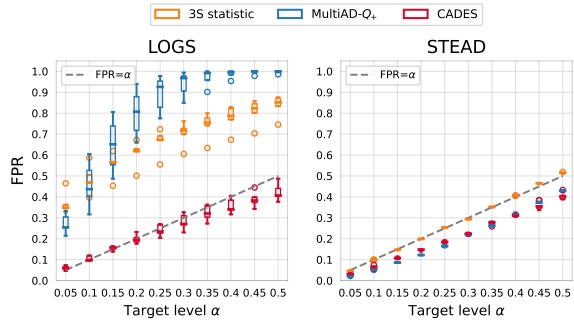

*Figure 4.* Boxplots of FPR for OOD detection on two real-world datasets under different target levels $\alpha$.

*Table 2.* TPR (%) for OOD detection on the STEAD dataset under the target level $\alpha = 0.05$.

| Dataset | 3S statistic | MultiAD-$Q_+$ | CADES |
|---|---|---|---|
| STEAD - Anchorage, AK | 74.30 | 67.14 | **95.46** |
| STEAD - Aleutian Islands, AK | **100** | **100** | **100** |
| STEAD - Helmet, CA | 69.20 | 6.50 | **98.38** |

## 4.5. Ablation Study

In this subsection, we conduct ablation studies to validate the effectiveness of combining two non-conformity scores and using two-sided p-values. Specifically, we consider

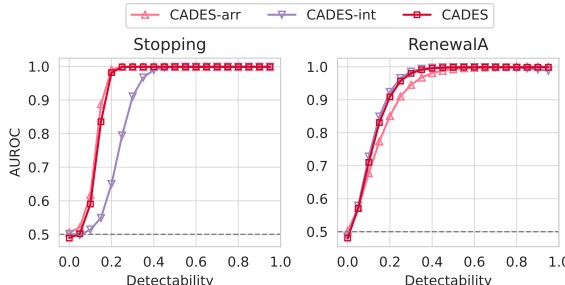

*Figure 5.* Performance of CADES-arr and CADES-int for the GOF test in the Stopping and RenewalA scenarios.

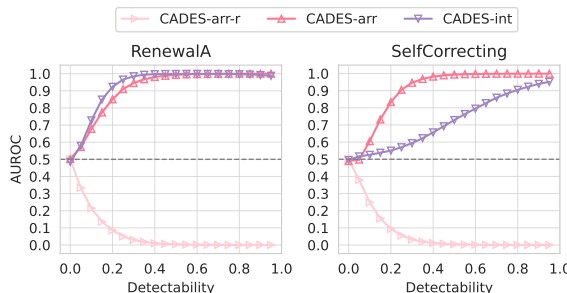

*Figure 6.* Performance of CADES-arr-r for the GOF test in the RenewalA and SelfCorrecting scenarios.

three model variants: (1) a single score $s_{\text{arr}}$ with the two-sided p-value, (2) a single score $s_{\text{int}}$ with the two-sided p-value, and (3) a single score $s_{\text{arr}}$ with the right-sided p-value. We denote these three variants as CADES-arr, CADES-int, and CADES-arr-r, respectively.

We evaluate CADES-arr and CADES-int in the Stopping and RenewalA scenarios. As shown in Figure 5, CADES-arr achieves strong performance in the Stopping scenario, where the number of events in the OOD sequence decreases, while CADES-int excels in the RenewalA scenario, where the spacings in the OOD sequence are relatively uniform. These successes can be attributed to the sensitivity of $s_{\text{arr}}$ to reductions in event count and the sensitivity of $s_{\text{int}}$ to relatively uniform spacings. By contrast, CADES, which integrates these two scores, demonstrates superior overall performance across both scenarios. We also conduct similar experiments on real-world data, with the results provided in Appendix D.3. These findings further validate the advantages of using $s_{\text{arr}}$ and $s_{\text{int}}$ simultaneously.

For CADES-arr-r, we focus on the RenewalA and SelfCorrecting scenarios, where the spacings of the OOD sequence are more uniform than those of the ID sequence generated from the SPP. As shown in Figure 6, CADES-arr-r performs poorly in these two scenarios, with significant classification errors. We explain this as follows. The distributions of $s_{\text{arr}}$ in these two scenarios are presented in Figure 7. In both cases, smaller values of $s_{\text{arr}}$ tend to indicate OOD sequences. Consequently, the right-sided p-value in CADES-arr-r assigns

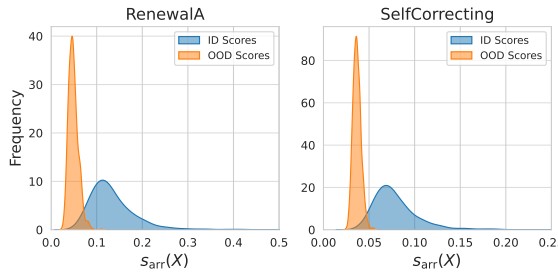

*Figure 7.* Distribution of the score $s_{\mathrm{arr}}(X)$ in the RenewalA and SelfCorrecting scenarios at the detectability $\eta = 0.5$.

larger p-values to OOD sequences than to ID sequences, resulting in incorrect classification. A similar analysis can be conducted in the Uniform scenario. Additionally, from Figure 6, we observe that CADES-arr, which uses the two-sided p-value, significantly outperforms CADES-int in the SelfCorrecting scenario. This also highlights the necessity of using the two-sided p-value.

# 5. Conclusion

We introduced CADES, a novel conformal anomaly detection method for continuous-time event sequences. CADES combines two newly designed non-conformity scores with provably valid p-values for hypothesis testing and provides theoretical guarantees on the calibration-conditional FPR. Extensive experiments demonstrate that CADES outperforms state-of-the-art methods in both GOF test for SPP and anomaly detection on synthetic and real-world datasets, while controlling the FPR at a pre-specified level.

## Acknowledgements

The authors would like to thank the anonymous reviewers and area chairs for their valuable comments and suggestions. This work was partially supported by the Strategic Priority Research Program of the Chinese Academy of Sciences (XDB0680101), the National Natural Science Foundation of China (62472416, 62376064, U2336202, 62402491), and the CAS Project for Young Scientists in Basic Research (YSBR-008).

## Impact Statement

This paper presents work whose goal is to advance the field of Machine Learning. There are many potential societal consequences of our work, none which we feel must be specifically highlighted here.

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

# A. Related Work

**Conformal Anomaly Detection.** Conformal anomaly detection (Smith et al., 2015; Laxhammar & Falkman, 2015; Ishimtsev et al., 2017), also referred to as conformal out-of-distribution (OOD) detection (Kaur et al., 2022; Lin et al., 2025), is an application of conformal inference/prediction (Vovk et al., 2005) in the context of anomaly detection or OOD detection. It employs a non-conformity score to quantify how different an input is from the training (in-distribution) data, and then computes the conformal p-value to assess the abnormality. Recent advances in this field have gained considerable attention (Kaur et al., 2022; Bates et al., 2023; Ma et al., 2024; Marandon et al., 2024; Liang et al., 2024; Lee et al., 2025). However, these works mainly focus on image data, and none have yet considered continuous-time event sequences. To fill this gap, we design powerful non-conformity scores tailored to event sequences and develop a valid test procedure based on the inductive conformal inference. Moreover, we provide calibration-conditional FPR guarantees (Magesh et al., 2023), which are stronger than marginal FPR guarantees (Kaur et al., 2022; Haroush et al., 2022; Lin et al., 2025).

**Anomaly Detection in Event Sequences.** Temporal point processes (TPPs) are probabilistic models for continuous-time event sequences (Zuo et al., 2020; Lin et al., 2021; Lüdke et al., 2023). With the development of neural networks, neural TPPs have become a promising method for modeling complex event dynamics (Li et al., 2024; Zhang et al., 2024a; Zeng et al., 2024; Yang et al., 2024). Naumzik & Feuerriegel (2022) developed a mixture marked Hawkes model to detect false rumors on social media. Liu & Hauskrecht (2021); Nath et al. (2024) focused on detecting specific anomalous events in a sequence using neural TPPs, whereas our work addresses the detection of entire anomalous sequences. The most relevant studies to ours are (Shchur et al., 2021; Zhang et al., 2023), which perform hypothesis testing to identify anomalous event sequences. Compared to these works, we address the non-identifiability issues of their test statistics and provide theoretical guarantees for both marginal FPR and calibration-conditional FPR.

**Goodness-of-Fit Test for TPPs.** Many popular goodness-of-fit (GOF) tests for TPPs rely on the time-rescaling theorem (Ogata, 1988; Brown et al., 2002; Gerhard et al., 2011; Tao et al., 2018), reducing the problem of GOF testing for an arbitrary TPP to determining whether the rescaled sequence follows the standard Poisson process (SPP). In practice, GOF tests for the SPP typically use the Kolmogorov-Smirnov (KS) statistic to check if the arrival times are distributed uniformly (Barnard, 1953) or if the inter-event times follow an exponential distribution (Cox & Lewis, 1966). However, the KS statistic performs poorly in scenarios involving changes in the event count (Shchur et al., 2021; Zhang et al., 2024b). More failure modes of KS statistics can be found in (Pillow, 2009). While the sum-of-squared-spacings (3S) statistic (Shchur et al., 2021) and the $Q_+$ and $Q_-$ statistics (Zhang et al., 2023) address some shortcomings of KS statistics, they are insensitive to relatively uniform spacings, which significantly reduces their statistical power. In this paper, we show that our two non-conformity scores capture complementary sensitivities to the event count and relatively uniform spacings. To make full use of their respective strengths, we apply the Bonferroni correction to combine these two scores for hypothesis testing.

**Conformal Prediction for Time Series.** Conformal prediction (CP) has gained traction in the field of time series forecasting (Stankeviciute et al., 2021; Angelopoulos et al., 2023; Auer et al., 2023), providing a powerful tool for uncertainty quantification. Traditionally, CP methods require exchangeability of data, a condition often violated by time series data due to inherent temporal dependencies. To address this challenge, various approaches have been developed to adapt CP for time series (Xu & Xie, 2021; Gibbs & Candes, 2021; Zaffran et al., 2022; Xu & Xie, 2023a;b). While both time series and event sequences are types of sequential data, they differ significantly (Xiao et al., 2017; Zhang et al., 2024a), with event sequences commonly modeled using TPPs. Recently, Dheur et al. (2024) employed CP and neural TPPs to predict event times and marks, offering valid coverage guarantees. In contrast, our work focuses on the application of CP for anomaly detection in event sequences.

# B. Proofs

### B.1. Proof of Proposition 3.3

*Proof.* We aim to show that the p-value $p_{cor}(X_{test})$ in Eq.(11) is stochastically larger than or equal to the uniform distribution under the null hypothesis $H_0$, i.e., $X_{test} \sim P_X$. Since the conformal p-values, such as $p^l_{arr}(X_{test})$ and $p^r_{arr}(X_{test})$, are valid under $H_0$ (Papadopoulos et al., 2002; Vovk et al., 2005), it follows that for every $\alpha \in (0,1)$,

$$\mathbb{P}_{H_0}\left(p^l_{arr}(X_{test}) \leq \alpha\right) \leq \alpha \quad \text{and} \quad \mathbb{P}_{H_0}\left(p^r_{arr}(X_{test}) \leq \alpha\right) \leq \alpha. \tag{15}$$

These inequalities follow from the exchangeability of the non-conformity scores $s_{arr}(X_{test})$ and $\{s_{arr}(X_{cal}) : X_{cal} \in \mathcal{D}_{cal}\}$.

Using the Union Bound, we obtain the following for every $\alpha \in (0, 1)$,

$$\mathbb{P}_{H_0}(p_{\mathrm{arr}}(X_{\mathrm{test}}) \leq \alpha) = \mathbb{P}_{H_0}\big(\min\{2p_{\mathrm{arr}}^l(X_{\mathrm{test}}),\, 2p_{\mathrm{arr}}^r(X_{\mathrm{test}})\} \leq \alpha\big) \tag{16}$$

$$= \mathbb{P}_{H_0}\big(\{2p_{\mathrm{arr}}^l(X_{\mathrm{test}}) \leq \alpha\} \cup \{2p_{\mathrm{arr}}^r(X_{\mathrm{test}}) \leq \alpha\}\big) \tag{17}$$

$$\leq \mathbb{P}_{H_0}\big(2p_{\mathrm{arr}}^l(X_{\mathrm{test}}) \leq \alpha\big) + \mathbb{P}_{H_0}\big(2p_{\mathrm{arr}}^r(X_{\mathrm{test}}) \leq \alpha\big) \tag{18}$$

$$\leq \frac{\alpha}{2} + \frac{\alpha}{2} \tag{19}$$

$$= \alpha. \tag{20}$$

Similarly, we also have $\mathbb{P}_{H_0}(p_{\mathrm{int}}(X_{\mathrm{test}}) \leq \alpha) \leq \alpha$. Then, for every $\alpha \in (0, 1)$ and $\varepsilon \geq 0$, we obtain:

$$\mathbb{P}_{H_0}(p_{\mathrm{cor}}(X_{\mathrm{test}}) \leq \alpha) = \mathbb{P}_{H_0}\big(\min\{2(1+\varepsilon)p_{\mathrm{arr}}(X_{\mathrm{test}}),\, 2(1+\varepsilon)p_{\mathrm{int}}(X_{\mathrm{test}})\} \leq \alpha\big) \tag{21}$$

$$\leq \mathbb{P}_{H_0}\Big(p_{\mathrm{arr}}(X_{\mathrm{test}}) \leq \frac{\alpha}{2(1+\varepsilon)}\Big) + \mathbb{P}_{H_0}\Big(p_{\mathrm{int}}(X_{\mathrm{test}}) \leq \frac{\alpha}{2(1+\varepsilon)}\Big) \tag{22}$$

$$\leq \frac{\alpha}{2(1+\varepsilon)} + \frac{\alpha}{2(1+\varepsilon)} \tag{23}$$

$$\leq \alpha. \tag{24}$$

This completes the proof. $\qquad\square$

## B.2. Proof of Theorem 3.4

*Proof.* According to Algorithm 1, the probability of declaring that $X_{\mathrm{test}}$ is OOD, conditioned on the calibration set $\mathcal{D}_{\mathrm{cal}}$, is given by:

$$\mathbb{P}_{H_0}(\text{declare OOD} \mid \mathcal{D}_{\mathrm{cal}}) = \mathbb{P}_{H_0}\big(p_{\mathrm{cor}}(X_{\mathrm{test}}) \leq \alpha \mid \mathcal{D}_{\mathrm{cal}}\big). \tag{25}$$

From the Union Bound, we derive

$$\mathbb{P}_{H_0}\big(p_{\mathrm{cor}}(X_{\mathrm{test}}) \leq \alpha \mid \mathcal{D}_{\mathrm{cal}}\big) = \mathbb{P}_{H_0}\big(\min\{2(1+\varepsilon)p_{\mathrm{arr}}(X_{\mathrm{test}}),\, 2(1+\varepsilon)p_{\mathrm{int}}(X_{\mathrm{test}})\} \leq \alpha \mid \mathcal{D}_{\mathrm{cal}}\big) \tag{26}$$

$$= \mathbb{P}_{H_0}\Big(\bigcup_{i=1}^{2}\Big\{p_i(X_{\mathrm{test}}) \leq \frac{\alpha}{2(1+\varepsilon)}\Big\} \mid \mathcal{D}_{\mathrm{cal}}\Big) \tag{27}$$

$$\leq \sum_{i=1}^{2}\mathbb{P}_{H_0}\Big(p_i(X_{\mathrm{test}}) \leq \frac{\alpha}{2(1+\varepsilon)} \mid \mathcal{D}_{\mathrm{cal}}\Big) \tag{28}$$

$$\leq \sum_{i=1}^{2}\sum_{j=1}^{2}\mathbb{P}_{H_0}\Big(p_i^j(X_{\mathrm{test}}) \leq \frac{\alpha}{4(1+\varepsilon)} \mid \mathcal{D}_{\mathrm{cal}}\Big) \tag{29}$$

$$= \sum_{i=1}^{2}\sum_{j=1}^{2} r_i^j. \tag{30}$$

Here, we use the notation $i \in \{1, 2\}$ to represent "arr" and "int", and $j \in \{1, 2\}$ to represent "$l$" and "$r$" for convenience. We denote $r_i^j = \mathbb{P}_{H_0}\big(p_i^j(X_{\mathrm{test}}) \leq \frac{\alpha}{4(1+\varepsilon)} \mid \mathcal{D}_{\mathrm{cal}}\big)$ for $i, j \in \{1, 2\}$.

When the non-conformity score follows a continuous distribution, the cumulative distribution function (CDF) of the conformal p-value conditioned on the calibration set follows a Beta distribution (Vovk, 2012; Bates et al., 2023). Due to symmetry, this property holds for both the left-sided and right-sided conformal p-values. Specifically, for each $r_i^j$, we have $r_i^j \sim \mathrm{Beta}(a, b)$, where $a = \big\lfloor (n_{\mathrm{cal}} + 1)\frac{\alpha}{4(1+\varepsilon)}\big\rfloor$ and $b = n_{\mathrm{cal}} + 1 - a$. The mean of this distribution is $\mu = \frac{a}{a+b}$.

Let $E$ represent the event

$$E = \bigcap_{i=1}^{2}\bigcap_{j=1}^{2}\Big\{r_i^j \leq \frac{\alpha}{4}\Big\}. \tag{31}$$

Then,

$$1 - \mathbb{P}(E) = 1 - \mathbb{P}\Big(\bigcap_{i=1}^{2}\bigcap_{j=1}^{2}\Big\{r_i^j \leq \frac{\alpha}{4}\Big\}\Big) \leq \sum_{i=1}^{2}\sum_{j=1}^{2}\mathbb{P}\Big(r_i^j \geq \frac{\alpha}{4}\Big) = \sum_{i=1}^{2}\sum_{j=1}^{2} 1 - I_{\frac{\alpha}{4}}(a, b), \tag{32}$$

where $I_x(a, b)$ denotes the CDF of the $\text{Beta}(a, b)$ distribution. Since $n_{\text{cal}}$ satisfies the condition in (13) and $\mu$ is upper bounded by $\frac{\alpha}{4(1+\varepsilon)}$, we obtain

$$1 - I_{\frac{\alpha}{4}}(a, b) \leq 1 - I_{(1+\varepsilon)\mu}(a, b) \leq \frac{\delta}{4}. \tag{33}$$

Thus, it follows that

$$1 - \mathbb{P}(E) \leq \sum_{i=1}^{2} \sum_{j=1}^{2} \frac{\delta}{4} = \delta. \tag{34}$$

Then, under event $E$, i.e., with probability at least $1 - \delta$, we have

$$\mathbb{P}_{H_0}(\text{declare OOD} \mid \mathcal{D}_{\text{cal}}) \leq \sum_{i=1}^{2} \sum_{j=1}^{2} r_i^j \leq \sum_{i=1}^{2} \sum_{j=1}^{2} \frac{\alpha}{4} = \alpha. \tag{35}$$

This completes the proof. ☐

## C. Experimental Details

### C.1. Dataset Descriptions

#### C.1.1. ALTERNATIVE DISTRIBUTIONS

In Section 4.1, we consider the following nine alternative distributions, where $\eta \in [0, 1]$ is the detectability parameter.

- **DecreasingRate:** Homogeneous Poisson process with intensity $\lambda = 1 - 0.5\eta$.

- **IncreasingRate:** Homogeneous Poisson process with intensity $\lambda = 1 + 0.5\eta$.

- **InhomogeneousPoisson:** Inhomogeneous Poisson process with intensity $\lambda(t) = 1 + \beta \sin(\omega t)$, where $\omega = \frac{2\pi}{50}$ and $\beta = 2\eta$.

- **Stopping:** Events occurring in $[t_{\text{stop}}, T]$ are removed from the SPP sequence, where $t_{\text{stop}} = T(1 - 0.3\eta)$ and $T = 100$.

- **RenewalA:** A renewal process, where inter-event times $\tau_i$ are sampled i.i.d. from a Gamma distribution with shape $k = \frac{1}{1-\eta}$ and scale $\theta = 1 - \eta$. In this case, $\mathbb{E}[\tau_i] = k\theta = 1$ and $\text{Var}[\tau_i] = k\theta^2 = 1 - \eta$. Thus, the expected inter-event time remains constant at 1, but the variance of inter-event times decreases for higher $\eta$.

- **RenewalB:** A renewal process, where inter-event times $\tau_i$ are sampled i.i.d. from a Gamma distribution with shape $k = 1 - \eta$ and scale $\theta = \frac{1}{1-\eta}$. The expected inter-event time remains constant, but the variance increases for higher $\eta$.

- **Hawkes:** Hawkes process with intensity $\lambda^*(t) = \mu + \alpha \sum_{t_i < t} \exp(-(t - t_i))$, where $\mu = 1 - \eta$ and $\alpha = \eta$.

- **SelfCorrecting:** Self-correcting process with intensity $\lambda^*(t) = \exp(\mu t - \sum_{t_i < t} \alpha)$, where $\mu = \eta + 10^{-5}$ and $\alpha = \eta$.

- **Uniform:** The inter-event times $\tau_i$ in a sequence are all $1 + 2\eta$, i.e., $\tau_i = 1 + 2\eta$.

#### C.1.2. SYNTHETIC DATASETS

We provide an overview of four synthetic datasets and two real-world datasets below. More detailed descriptions can be found in the original paper (Shchur et al., 2021). Table 3 summarizes statistics of these datasets.

- **ServerStop** and **ServerOverload:** ID sequences for both scenarios are generated by a multivariate Hawkes process with $M = 3$ marks on the interval $[0, 100]$, modeling network traffic among 3 hosts. In OOD sequences, the influence matrix is changed to simulate two scenarios: (1) a host goes offline (**ServerStop**), and (2) a host goes down and the traffic is routed to another host (**ServerOverload**). A higher detectability $\eta$ indicates that the change in the influence matrix occurs earlier.

*Table 3.* Statistics of synthetic and real-world datasets.

| Dataset | # Marks | $|\mathcal{D}|$ | $|\mathcal{D}^{\mathrm{ID}}_{\mathrm{test}}|$ | $|\mathcal{D}^{\mathrm{OOD}}_{\mathrm{test}}|$ |
|---|---|---|---|---|
| ServerStop | 3 | 1000 | 1000 | 1000 |
| ServerOverload | 3 | 1000 | 1000 | 1000 |
| Latency | 2 | 1000 | 1000 | 1000 |
| SpikeTrains | 50 | 500 | 96 | 96 |
| LOGS | 8 | 1668 | 502 | 110 |
| STEAD | 1 | 4000 | 1000 | 3000 |

- **Latency:** Each sequence contains two marks on the interval $[0, 100]$. In ID sequences, the first mark, the "trigger", is generated by a homogeneous Poisson process with intensity $\lambda = 3$. The second mark, the "response", is obtained by shifting the arrival times of the first mark by offsets, which are independently sampled from $\mathrm{Normal}(\mu = 1, \sigma = 0.1)$. In OOD sequences, the offsets are instead sampled from $\mathrm{Normal}(\mu = 1 + 0.5\eta, \sigma = 0.1)$, introducing an increased latency between the "trigger" and "response" events.

- **SpikeTrains:** ID sequences consist of the firing times of 50 neurons observed over a 20-second period, with each neuron represented by a distinct mark. OOD sequences are obtained by switching event marks (e.g., switching marks 1 and 2). A higher detectability $\eta$ corresponds to a greater number of switches.

### C.1.3. REAL-WORLD DATASETS

- **LOGS:** Event sequences represent the timestamps of 8 types of log entries. Each type of log entry is treated as a mark, and each sequence is observed in 30 seconds. ID sequences correspond to normal operations, while OOD sequences are generated by injecting failures using a chaos testing tool Pumba. OOD sequences contain 5 types of injected anomalies (e.g., packet corruption, increased latency). See Table 1 for the list of anomalies. $\mathcal{D}$, $\mathcal{D}^{\mathrm{ID}}_{\mathrm{test}}$, and $\mathcal{D}^{\mathrm{OOD}}_{\mathrm{test}}$ consist of 1668 ID sequences, 502 ID sequences, and 22 OOD sequences per each failure injection scenario, respectively.

- **STEAD** (Stanford Earthquake Dataset): Event sequences contain the occurrence times of earthquakes within a 350km radius of 4 geographical locations: San Mateo, CA; Anchorage, AK; Aleutian Islands, AK; and Helmet, CA. Each sequence is unmarked and observed over 72 hours. The sequences corresponding to San Mateo, CA, are used as ID data, and the remaining 3 locations as OOD data. $\mathcal{D}$, $\mathcal{D}^{\mathrm{ID}}_{\mathrm{test}}$, and $\mathcal{D}^{\mathrm{OOD}}_{\mathrm{test}}$ consist of 4000 ID sequences, 1000 ID sequences, and 1000 OOD sequences per each remaining location, respectively.

To ensure a fair evaluation of the anomaly detection performance, we excluded three normal (ID) sequences with the shortest lengths (0, 120, and 196) from the test set of the LOGS dataset. These sequences are significantly shorter than the average length of approximately 5000 for normal sequences and are therefore not representative of typical normal data patterns.

### C.2. Test Statistics

Let $Z = (\tau_1, \ldots, \tau_N)$ be an unmarked event sequence on the interval $[0, V]$, with $\tau_0 = 0$ and $\tau_{N+1} = V$. Let $w_i = \tau_i - \tau_{i-1}$ for $i = 1, \ldots, N + 1$. The test statistics, used as baseline methods in this paper, are provided below.

- **KS arrival:**

$$\kappa_{\mathrm{arr}}(Z) = \sqrt{N} \sup_{\tau \in [0,V]} \left| \hat{F}_{\mathrm{arr}}(\tau) - F_{\mathrm{arr}}(\tau) \right|,$$

where $\hat{F}_{\mathrm{arr}}(\tau) = \frac{1}{N} \sum_{i=1}^{N} \mathbb{1}_{[\tau_i, \infty)}(\tau)$ and $F_{\mathrm{arr}}(\tau) = \tau/V$.

- **KS inter-event:**

$$\kappa_{\mathrm{int}}(Z) = \sqrt{N} \sup_{w \in [0,\infty)} \left| \hat{F}_{\mathrm{int}}(w) - F_{\mathrm{int}}(w) \right|,$$

where $\hat{F}_{\mathrm{int}}(w) = \frac{1}{N+1} \sum_{i=1}^{N+1} \mathbb{1}_{[w_i, \infty)}(w)$ and $F_{\mathrm{int}}(w) = 1 - \exp(-w)$.

- **Chi-squared:** Following Shchur et al. (2021), the interval $[0, V]$ is partitioned into $B = 10$ disjoint buckets of equal length, and then the observed event count $N_b$ in each bucket is compared with the expected amount $L = V/B$. Thus, the Chi-squared statistic is defined as

$$\chi^2(Z) = \sum_{b=1}^{B} \frac{(N_b - L)^2}{L}.$$

- **Sum-of-squared-spacings (3S) statistic:**

$$\psi(Z) = \frac{1}{V} \sum_{i=1}^{N+1} w_i^2.$$

- **$Q_+$ statistic:**

$$Q_+(Z) = \frac{1}{V} \Big( \sum_{i=1}^{N+1} w_i^2 + \sum_{i=1}^{N} w_i w_{i+1} \Big).$$

- **$Q_-$ statistic:**

$$Q_-(Z) = \frac{1}{V} \Big( \sum_{i=1}^{N+1} w_i^2 - \sum_{i=1}^{N} w_i w_{i+1} \Big).$$

- **Log-likelihood:** Different from the above statistics, the log-likelihood statistic (used in Sections 4.2 and 4.3) is computed directly on the marked event sequence using the condition intensity function of the learned neural TPP model. Specifically, the log-likelihood function of a marked event sequence $X = \{(t_i, m_i)\}_{i=1}^{N}$ on the interval $[0, T]$ is defined as follows:

$$\mathcal{L}(X) = \sum_{i=1}^{N} \log \lambda_{m_i}^*(t_i) - \sum_{m=1}^{M} \Big( \int_0^T \lambda_m^*(s) \, \mathrm{d}s \Big).$$

### C.3. Training Details

In the case of the GOF test for SPP (Section 4.1), there is no need to train a neural TPP model, as the distribution under the null hypothesis is known. In contrast, for OOD detection (Sections 4.2 and 4.3), where the distribution under the null hypothesis is unknown, we train a neural TPP (LogNormMix, Shchur et al. (2020)) using the same model architecture as the baselines (Shchur et al., 2021; Zhang et al., 2023). For completeness, we reiterate the details here. The inter-event time distribution is parameterized with a mixture of 8 Weibull distributions, the mark embedding size is set to 32, and the RNN hidden size is set to 64 for all experiments.

We randomly split the dataset $\mathcal{D}$ into two disjoint subsets of equal size: a training set $\mathcal{D}_{\text{train}}$ and a calibration set $\mathcal{D}_{\text{cal}}$. Unlike the baselines that optimize the model parameters by maximizing the log-likelihood of the sequences in the entire dataset $\mathcal{D}$, we train the model using the sequences in $\mathcal{D}_{\text{train}}$. The following training procedure is the same for both our method and the baselines. The batch size is set to 64, the optimizer is Adam with a learning rate of $10^{-3}$, and the $L_2$ norm of the gradient is clipped to 5. We set the maximum number of epochs to 500 and perform early stopping if the training loss does not improve for 5 epochs.

### C.4. Implementation Details

All experiments in this paper are conducted on an NVIDIA RTX 3090 Ti GPU using PyTorch. For kernel density estimation (KDE) in our test procedure, we use "scipy.stats.gaussian_kde" with the parameter "bw_method" set to $h_1$ for $s_{\text{arr}}(X)$ and $h_2$ for $s_{\text{int}}(X)$. The values of $h_1$ and $h_2$ are selected from the grid $\{0.001, 0.005, 0.01, 0.05, 0.1, 0.5, 1.0\}$. Following the baselines, the results reported in Section 4.1 and Section 4.2 are averaged over 10 random seeds. In Section 4.3, we train the neural TPP model with 5 different random initializations to compute the average results.

## D. Extra Experimental Results

### D.1. Runtime Comparison

We compare the inference runtime of CADES with the baselines on two real-world datasets. The experimental results, averaged over five runs, are presented in Table 4. CADES requires slightly longer inference times compared to the baselines.

The additional time taken by CADES can be attributed to its use of two non-conformity scores simultaneously and the need for numerical integration. Despite this slight increase in runtime, the results indicate that the computational overhead introduced by CADES is acceptable and does not diverge significantly from the baselines.

*Table 4.* Inference runtime averaged over 5 runs (in seconds).

| Dataset | 3S statistic | MultiAD-$Q_+$ | CADES |
|---------|--------------|---------------|-------|
| LOGS | 24.49 | 30.36 | 38.32 |
| STEAD | 19.14 | 22.27 | 25.98 |

### D.2. Additional GOF Test for SPP

The AUROC scores for the IncreasingRate, InhomogeneousPoisson, and RenewalB scenarios are shown in Figure 8. These results are consistent with the conclusions presented in Section 4.1. In these cases, CADES achieves superior or comparable performance, validating its effectiveness across patterns distinct from the SPP.

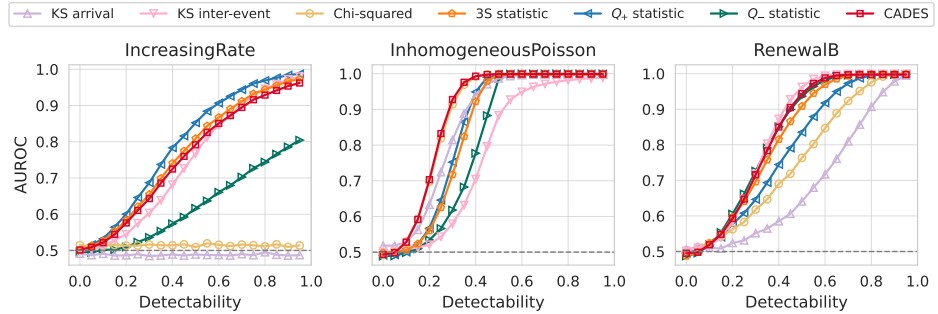

*Figure 8.* Performance of GOF test for the SPP under other three scenarios.

### D.3. Ablation Study on Real-World Data

The results of CADES-arr and CADES-int on two real-world datasets are presented in Table 5. CADES-arr and CADES-int refer to the variants that use a single non-conformity score $s_{\mathrm{arr}}(X)$ and $s_{\mathrm{int}}(X)$, respectively. We can observe that CADES achieves superior overall performance. Notably, in certain scenarios, both CADES-arr and CADES-int perform worse than CADES, or one of them performs poorly such as CADES-arr in the "LOGS - Packet duplication (1%)" scenario. These observations highlight the benefits of using $s_{\mathrm{arr}}(X)$ and $s_{\mathrm{int}}(X)$ simultaneously.

*Table 5.* AUROC (%) results of the CADES variants for OOD detection on real-world datasets.

| Dataset | CADES-arr | CADES-int | CADES |
|---------|-----------|-----------|-------|
| LOGS - Packet corruption (1%) | 94.14 | 90.59 | **96.48** |
| LOGS - Packet corruption (10%) | **99.60** | 97.84 | 99.48 |
| LOGS - Packet duplication (1%) | 58.43 | **92.98** | 92.88 |
| LOGS - Packet delay (frontend) | **98.24** | 96.99 | 98.15 |
| LOGS - Packet delay (all services) | 98.22 | 99.24 | **99.33** |
| STEAD - Anchorage, AK | 84.60 | 98.92 | **99.31** |
| STEAD - Aleutian Islands, AK | 99.85 | 99.82 | **99.95** |
| STEAD - Helmet, CA | 82.67 | **99.67** | 99.30 |
| Average Rank | 2.25 | 2.25 | **1.50** |

