# OpenReview forum: "Conformal Anomaly Detection in Event Sequences"
_ICML.cc/2025/Conference — ICML 2025 poster_

### Official Review · Reviewer_aGKQ · 2025-02-24

**Overall Recommendation:** 5

**Summary:**

The paper introduces CADES, a novel anomaly detection method for continuous-time event sequences under the conformal inference framework. The authors propose two new non-conformity scores tailored to event sequences based on the time-rescaling theorem, which address the non-identifiability issues of existing test statistics. CADES combines these scores with Bonferroni correction to conduct statistical hypothesis testing and provides theoretical guarantees on the false positive rate (FPR). Extensive experiments on synthetic and real-world datasets demonstrate that CADES outperforms state-of-the-art methods while maintaining FPR control.

**Claims And Evidence:**

The claims are well-supported by both theoretical guarantees and empirical results.

- The paper provides rigorous proofs for the validity of the p-values used in CADES and offers theoretical guarantees on calibration-conditional FPR control.

- The experimental results show that CADES achieves superior detection performance compared to state-of-the-art methods while controlling the FPR at a pre-specified level.

**Essential References Not Discussed:**

No. The paper covers the essential references related to conformal inference, event sequence anomaly detection, and temporal point processes.

**Experimental Designs Or Analyses:**

The experimental designs and analyses are sound and well-executed. It is worth noting that the summary and visualization of the experimental results are neat and clear.

- The paper conducts experiments on synthetic datasets to evaluate CADES' performance in GOF testing for the SPP and addressing the non-identifiability issues, as well as on real-world datasets to demonstrate its practical applicability in detecting various types of anomalies and controlling the FPR.

- The ablation studies verify the effectiveness of combining two non-conformity scores and using two-sided p-values.

**Methods And Evaluation Criteria:**

The proposed methods and evaluation criteria make sense for the problem of anomaly detection in event sequences.

- The application of conformal inference and the design of two non-conformity scores effectively address the limitations of existing methods, particularly their inability to control the FPR and the non-identifiability issues.

- The evaluation criteria on synthetic and real-world datasets, including AUROC, FPR control, and runtime comparisons, are appropriate for assessing the performance of anomaly detection methods.

**Other Comments Or Suggestions:**

Suggestions:

- The paper could explore the potential for extending the proposed method to other types of sequential data, such as time series or spatial-temporal data, to broaden its applicability.

- It would be clearer to denote the new observation $X$ in the problem statement of Section 2 as $X_{\text{test}}$.

**Other Strengths And Weaknesses:**

Strengths:

- **Originality**: The paper establishes a novel connection between conformal inference and anomaly detection in event sequences. It combines two newly designed non-conformity scores with Bonferroni correction to conduct statistical hypothesis testing and addresses the non-identifiability issues of existing approaches.

- **Theoretical Analysis**: The paper offers theoretical support for FPR control of anomaly detection in event sequences. Furthermore, the paper provides guarantees on calibration-conditional FPR control, which are stronger than the marginal FPR guarantees.

- **Evaluation**: The experimental evaluations are thorough and well-designed. The results demonstrate that CADES outperforms state-of-the-art methods on both synthetic and real-world datasets while effectively controlling the FPR at a pre-specified level.

- **Presentation**: The paper is well-organized. It presents a clear explanation of the motivation, methods, and experiments. This makes the paper easy to follow.

Weaknesses:

- While the authors provide theoretical guarantees on FPR control, the practical implication and significance of these guarantees in real-world scenarios could be explained in more detail.

**Questions For Authors:**

Please refer to weaknesses and suggestions.

**Relation To Broader Scientific Literature:**

The key contributions of this paper lie in its exploration of conformal inference and anomaly detection in event sequences. The paper effectively bridges the gap between these two domains, making a significant advancement, which could also inspire future work on anomaly detection in other types of sequential data.

**Theoretical Claims:**

The theoretical claims about the validity of the p-values used in CADES and the guarantees on calibration-conditional FPR are backed by rigorous proofs. The authors also provide additional theoretical insights into the conditions under which these guarantees hold, especially with regard to the size of the calibration set.

---

> ### Author Rebuttal · Authors · 2025-03-30
>
> Thank you for your appreciation of our work and suggestions. We provide answers to your concerns as follows:
>
> **Q1**: While the authors provide theoretical guarantees on FPR control, the practical implication and significance of these guarantees in real-world scenarios could be explained in more detail.
>
> **R1**: FPR control guarantees are crucial for safety-critical applications, such as cybersecurity, finance, and healthcare. For example, in electronic health records (EHRs), maintaining low false positive rates is vital. High false positives could lead to incorrect diagnoses or unnecessary treatments, potentially compromising patient safety. Our method ensures that detected anomalies are more likely to be genuine, helping healthcare professionals make more reliable decisions. We will add this content in our paper.
>
> **Q2**: The paper could explore the potential for extending the proposed method to other types of sequential data, such as time series or spatial-temporal data, to broaden its applicability.
>
> **R2**: In this work, we primarily focus on anomaly detection in event sequences. Unlike other types of sequential data, event sequence data is asynchronous and of variable length, commonly modeled using temporal point processes. As you suggested, we plan to extend our method to other types of sequential data as future work.
>
> **Q3**: It would be clearer to denote the new observation $X$ in the problem statement of Section 2 as $X_{\text{test}}$.
>
> **R3**: Thank you for pointing this out. We have already made the replacement in our manuscript.

---

### Official Review · Reviewer_A6ji · 2025-03-03

**Overall Recommendation:** 4

**Summary:**

The paper is the first to extend conformal inference to anomaly detection in event sequences and proposes a novel method called CADES, which provides statistical guarantees of validity. Notably, CADES addresses the severe non-identifiability issues found in previous methods by developing two new powerful non-conformity scores. The authors provides both rigorous theoretical analysis of FPR and thorough evaluations, demonstrating CADES' strong performance in event sequence anomaly detection.

**Claims And Evidence:**

The claims are supported by rigorous theoretical analysis and superior empirical performance on both synthetic and real-world datasets.

**Essential References Not Discussed:**

While the paper cites relevant works well, it would be helpful to also discuss related work on applying conformal inference to time series (see Weaknesses for detailed references).

**Experimental Designs Or Analyses:**

The experiments are well-designed and the results show that CADES outperforms existing methods on both synthetic and real-world datasets. The ablation study further supports the importance of using the two proposed non-conformity scores and two-sided p-values.

**Methods And Evaluation Criteria:**

The proposed method is built on a statistically sound framework (conformal inference), and the evaluation criteria used (AUROC, TPR and FPR) are appropriate for the anomaly detection task.

**Other Comments Or Suggestions:**

No further comments or suggestions.

**Other Strengths And Weaknesses:**

Strengths:

- The idea of detecting anomalies in event sequences with conformal inference is novel and interesting. The paper is well-written and the presentation is coherent.

- The motivation is clear. This paper provides theoretical guarantees and empirical validation for FPR control, and proposes new non-conformity scores to address previous non-identifiability issues.

- The effectiveness of CADES is thoroughly evaluated on experiments, showcasing superior performance compared to existing approaches.

Weaknesses:

I don't find any obvious weaknesses. It would be more comprehensive if the literatures on applying conformal inference to time series were discussed.

[1] Xu C, Xie Y. Conformal prediction interval for dynamic time-series. ICML, 2021.

[2] Zaffran M, Féron O, Goude Y, et al. Adaptive conformal predictions for time series. ICML, 2022.

**Questions For Authors:**

What would the experimental performance be like if the KL divergence in Eq.(7) and Eq.(8) is replaced by the $L_2$ norm?

**Relation To Broader Scientific Literature:**

The paper clearly explains how it is related to previous work. The paper draws a new connection between conformal inference and event sequence anomaly detection, and then proposes an effective detection method with statistical guarantees.

**Theoretical Claims:**

I have checked the correctness of all proofs supporting the theoretical claims, which are both clear and solid.

---

> ### Author Rebuttal · Authors · 2025-03-30
>
> Thank you for your recognition and feedback. We provide answers to your concerns as follows:
>
> **Q1**: It would be more comprehensive if the literatures on applying conformal inference to time series were discussed.
>
> **R1**: Thank you for your valuable suggestion and the references you provided. We will incorporate a discussion of them in the paper. In this work, we specifically apply conformal inference to event sequences. Although both time series and event sequences are types of sequential data, they differ significantly [1]. Specifically, in time series, time serves only as the index to order the sequence of values for the target variable. In event sequences, time is treated as a random variable representing the timestamps of asynchronous events, commonly modeled using temporal point processes. In addition, the references you mentioned primarily focus on prediction tasks, while our work addresses the anomaly detection task.
>
> **Q2**: What would the experimental performance be like if the KL divergence in Eq.(7) and Eq.(8) is replaced by the $L_2$ norm?
>
> **R2**: We conducted experiments by replacing the KL divergence with the $L_2$ norm on real-world datasets, referring to this method as CADES-$L_2$. The results demonstrate that CADES with KL divergence outperforms CADES-$L_2$ in terms of AUROC.
>
> - AUROC (\%) results
>
> | Dataset | CADES-$L_2$ | CADES |
> | --- | --- | --- |
> | LOGS - Packet corruption (1\%) | 90.75 | 96.48 |
> | LOGS - Packet corruption (10\%) | 94.72 | 99.48 |
> | LOGS - Packet duplication (1\%) | 92.03 | 92.88 |
> | LOGS - Packet delay (frontend) | 93.66 | 98.15 |
> | LOGS - Packet delay (all services) | 93.49 | 99.33 |
> | STEAD - Anchorage, AK | 97.95 | 99.31 |
> | STEAD - Aleutian Islands, AK | 99.82 | 99.95 |
> | STEAD - Helmet, CA | 97.46 | 99.30 |
>
>
> Reference:
>
> [1] Xiao S, Yan J, Yang X, et al. Modeling the intensity function of point process via recurrent neural networks. AAAI, 2017.

---

### Official Review · Reviewer_T5Ha · 2025-03-13

**Overall Recommendation:** 4

**Summary:**

This paper proposes a novel test procedure based on conformal inference for detecting anomalous event sequences, with rigorous control over the false positive rate (FPR), a crucial factor for deploying anomaly detection methods in safety-critical applications. Specifically, it designs two new non-conformity scores tailored to event sequences, which capture complementary sensitivities to different abnormal patterns. By combining these scores with Bonferroni correction, the proposed method CADES overcomes the non-identifiability limitations of existing methods. Theoretically, this paper proves the validity of CADES and provides guarantees on calibration-conditional FPR. Experimental results validate the effectiveness of CADES across multiple benchmark datasets.

**Claims And Evidence:**

This paper provides clear and convincing evidence supporting its claims. The theoretical analysis of the proposed method, including guarantees on the marginal FPR and calibration-conditional FPR, is solid. The experiments demonstrate that CADES outperforms baseline methods, overcomes the non-identifiability limitations, and effectively controls FPR.

**Essential References Not Discussed:**

NA

**Experimental Designs Or Analyses:**

The experimental design is thorough, using multiple benchmark datasets with various anomaly types. The results highlight the superior performance of CADES compared to existing methods, both in terms of AUROC score and FPR control.

**Methods And Evaluation Criteria:**

The proposed method and evaluation criteria are reasonable and effective for detecting anomalous event sequences.

**Other Comments Or Suggestions:**

1. Could you provide the computational time of the CADES method?

2. How would CADES perform in scenarios where event sequences have sparse data?

**Other Strengths And Weaknesses:**

Strengths:

1. Applying conformal inference and neural temporal point processes to tackle the problem of anomaly detection in event sequences is both innovative and promising. This approach is backed by rigorous statistical theories, enhancing its reliability in real-world deployment.

2. The proposed two non-conformity scores, which overcome the non-identifiability limitations of existing test statistics, are well motivated and reasonable.

3. Extensive experiments across multiple benchmark datasets validate the effectiveness and reliability of the proposed method.

Weaknesses:

Some experimental results require further clarification, such as:

1. The reason behind the poor performance of the 3S statistic under the Uniform scenario in Section 4.1.

2. In Figure 7, why are the distributions of ID scores different in the RenewalA and SelfCorrecting scenarios?

**Questions For Authors:**

Please see above.

**Relation To Broader Scientific Literature:**

The authors successfully fill a gap in the literature by applying conformal inference to anomaly detection in event sequences. The two proposed non-conformity scores overcome the non-identifiability limitations of previous test statistics, thereby ensuring more accurate and robust detection results. The paper has the potential to inspire other researchers to further advance the field of anomaly detection.

**Theoretical Claims:**

The authors provide rigorous theoretical analysis, proving the validity of the proposed test procedure and ensuring calibration-conditional FPR control.

---

> ### Author Rebuttal · Authors · 2025-03-30
>
> Thank you for your insightful comments and questions. We provide answers to your concerns as follows:
>
> **Q1**: It is suggested to explain the reason behind the poor performance of the 3S statistic under the Uniform scenario in Section 4.1.
>
> **R1**: This is because the 3S statistic is not sensitive to relatively uniform spacings (i.e. inter-event times). Specifically, according to Proposition 1 in [1], the value of the 3S statistic for a standard Poisson process realization (i.e. an ID sequence) is around 2. In the Uniform scenario, the spacings of the OOD sequences are identical. For example, when the detectability parameter $\eta = 0.5$, the spacings are all 2. In this case, the 3S statistic for an OOD sequence is equal to 2, making it unable to distinguish between ID and OOD sequences. We will add this explanation in Section 4.1.
>
> **Q2**: In Figure 7, why are the distributions of ID scores different in the RenewalA and SelfCorrecting scenarios?
>
> **R2**: Since the bandwidth of the non-conformity score $s_{\text{arr}}(X)$ is selected differently in the RenewalA and SelfCorrecting scenarios, the value of $s_{\text{arr}}(X)$ is different, leading to differences in the distribution of ID scores.
>
> **Q3**: Could you provide the computational time of the CADES method?
>
> **R3**: During the training phase, for fairness, CADES employs the same neural TPP model as the baseline methods. Thus, its training time is similar to that of the baselines. We compare the inference runtimes of CADES and the baseline methods on real-world datasets in Appendix D.1, showing that CADES achieves comparable runtimes. For convenience, we also present the experimental results below:
>
> - Inference runtimes (in seconds)
>
> | Dataset | 3S statistic | MultiAD-$Q_+$ | CADES |
> | --- | --- | --- | --- |
> | LOGS | 24.49 | 30.36 | 38.32 |
> | STEAD | 19.14 | 22.27 | 25.98 |
>
> **Q4**: How would CADES perform in scenarios where event sequences have sparse data?
>
> **R4**: Our method is flexible, allowing any neural TPP model to be plugged into CADES. Therefore, we can use more advanced neural TPP models or models specifically designed for sparse event data to capture the distribution of the normal training data, and then apply the proposed test procedure for inference on the test data.
>
>
> Reference:
>
> [1] Shchur O, Turkmen A C, Januschowski T, et al. Detecting anomalous event sequences with temporal point processes. NeurIPS, 2021.

---

### Decision · Program_Chairs · 2025-05-01

**Decision:**

Accept (poster)

**Comment:**

This paper presents CADES, a method for anomaly detection in event sequences using conformal inference. The approach introduces non-conformity scores tailored to event data and combines them using Bonferroni correction to address challenges related to non-identifiability and FPR control. Reviewers agree that the paper is well-motivated, with theoretical analysis and comprehensive empirical validation on both synthetic and real-world datasets. The theoretical guarantees on calibration-conditional FPR and the ablation studies were noted as particular strengths. Some reviewers requested clarification on specific experimental results and suggested a broader discussion of related work on conformal inference in time series. The authors addressed these points in the rebuttal, providing additional explanations and experimental comparisons. While some aspects, such as the practical implications of FPR control, could be discussed in more detail, the overall assessment is that the paper makes a solid contribution to anomaly detection in sequential data.